# First-trimester exposure to benzodiazepines and risk of congenital malformations in offspring: A population-based cohort study in South Korea

**Yunha Noh**[1☯], **Hyesung Lee**[1,2☯], **Ahhyung Choi**[1], **Jun Soo Kwon**[3,4,5],
**Seung-Ah Choe**[6], **Jungmi Chae**[7], **Dong-Sook Kim**[7]*, **Ju-Young Shin**[1,2,8]*

1 School of Pharmacy, Sungkyunkwan University, Suwon, South Korea, 2 Department of Biohealth Regulatory Science, Sungkyunkwan University, Suwon, South Korea, 3 Department of Psychiatry, Seoul National University College of Medicine, South Korea, 4 Department of Brain and Cognitive Sciences, Seoul National University College of Natural Sciences, South Korea, 5 Institute of Human Behavioral Medicine, Seoul National University Medical Research Center, South Korea, 6 Division of Life Sciences, Korea University, Seoul, South Korea, 7 Department of Research, Health Insurance Review and Assessment Service, Wonju, South Korea, 8 Department of Clinical Research Design & Evaluation, Samsung Advanced Institute for Health Sciences & Technology (SAIHST), Sungkyunkwan University, Seoul, South Korea

☯ These authors contributed equally to this work.
* sttone@hira.or.kr (DSK); shin.jy@skku.edu (JYS)

**Data Availability Statement:** Data generated and/ or analyzed during the current study cannot be

## Abstract

### Background

Benzodiazepines are frequently prescribed during pregnancy; however, evidence about possible teratogenicity is equivocal. We aimed to evaluate the association between first-trimester benzodiazepine use and the risk of major congenital malformations.

### Methods and findings

Using Korea's nationwide healthcare database, we conducted a population-based cohort study of women who gave birth during 2011 to 2018 and their live-born infants. The exposure was defined as one or more benzodiazepine prescriptions during the first trimester. We determined the relative risks (RRs) and confidence intervals (CIs) of overall congenital malformations and 12 types of organ-specific malformations. Infants were followed from birth to death or 31 December 2019, whichever came first (up to 8 years of age). Propensity score fine stratification was employed to control for 45 potential confounders. Among a total of 3,094,227 pregnancies, 40,846 (1.3%) were exposed to benzodiazepines during the first trimester (mean [SD] age, 32.4 [4.1] years). The absolute risk of overall malformations was 65.3 per 1,000 pregnancies exposed to benzodiazepines versus 51.4 per 1,000 unexposed pregnancies. The adjusted RR was 1.09 (95% CI 1.05 to 1.13, $p < 0.001$) for overall malformations and 1.15 (1.10 to 1.21, $p < 0.001$) for heart defects. Based on mean daily lorazepam-equivalent doses, the adjusted RRs for overall malformations and heart defects were 1.05 (0.99 to 1.12, $p = 0.077$) and 1.12 (1.04 to 1.21, $p = 0.004$) for <1 mg/day and 1.26 (1.17 to 1.36, $p < 0.001$) and 1.31 (1.19 to 1.45, $p < 0.001$) for >2.5 mg/day doses,

shared publicly due to the data sharing policy of the Health Insurance Review and Assessment Service (HIRA) of Korea, governed by Article 18 of the Personal Information Protection Act ("Limitation to Out-of-Purpose Use and Provision of Personal Information" available at https://elaw.klri.re.kr/kor_service/lawView.do?hseq=53044&lang=ENG). However, the data are available from the HIRA on reasonable request for researchers who meet the criteria for access to confidential data (https://www.data.go.kr/en/tcs/eds/selectCoreDataView.do?coreDataInsttCode=B551182&coreDataSn=1&searchCondition2=coreDataNmEn&searchKeyword2=).

**Funding:** This work was supported by the National Research Foundation of Korea (NRF) grant funded by the Korean government (Ministry of Science and Information & Communication Technology, MSIT) (No. NRF-2020R1C1C1003527) (to J-YS); by Basic Science Research Program through the National Research Foundation of Korea (NRF) funded by the Ministry of Education (No. NRF-2021R1A6A3A13046424) (to YN) and by a grant (21153MFDS607) from Ministry of Food and Drug Safety of South Korea in 2021-2025 (to J-YS). The funders had no role in the study design, data collection and analysis, interpretation of data, writing of the report, and the decision to submit the article for publication.

**Competing interests:** We have read the journal's policy and the authors of this manuscript have the following competing interests: YN received a grant from the National Research Foundation of Korea, outside the submitted work. J-YS received grants from the Ministry of Food and Drug Safety, the Ministry of Health and Welfare, the National Research Foundation of Korea, the Government-wide R&D Fund for Infectious Disease Research and pharmaceutical companies, including Amgen, Pfizer, GSK, Hoffmann-La Roche, Dong-A ST, and Yungjin, outside the submitted work.

**Abbreviations:** aSD, absolute standardized difference; CI, confidence interval; HIRA, Health Insurance Review and Assessment Service; PAF, population attributable fraction; PS, propensity score; RR, relative risk; SD, standard deviation.

respectively, suggesting a dose–response relationship. A small but significant increase in risk for overall and heart defects was detected with several specific agents (range of adjusted RRs: 1.08 to 2.43). The findings were robust across all sensitivity analyses, and negative control analyses revealed a null association. Study limitations include possible exposure misclassification, residual confounding, and restriction to live births.

## Conclusions

In this large nationwide cohort study, we found that first-trimester benzodiazepine exposure was associated with a small increased risk of overall malformations and heart defects, particularly at the higher daily dose. The absolute risks and population attributable fractions were modest. The benefits of benzodiazepines for their major indications must be considered despite the potential risks; if their use is necessary, the lowest effective dosage should be prescribed to minimize the risk.

## Trial registration

ClinicalTrials.gov NCT04856436.

---

## Author summary

### Why was this study done?

- Anxiety and insomnia are common during pregnancy, and benzodiazepines are frequently prescribed for managing these conditions.

- The safety of benzodiazepines during pregnancy remains uncertain, as their evidence from epidemiological studies is limited and conflicting.

### What did the researchers do and find?

- In this large nationwide cohort study of more than 3 million pregnancies, we found a small increased risk of overall and heart defects associated with first-trimester benzodiazepine use.

- The risk of overall and heart defects was slightly increased at the high daily dose group, suggesting the dose–response relationship.

- A small but significant increased risk for overall and heart defects was detected with several specific benzodiazepines.

### What do these findings mean?

- The findings suggest that, although small, the potential risks should be evaluated against the efficacy of benzodiazepines and the lowest effective dosage should be recommended when prescribed in early pregnancy.

- The increased risk for congenital malformations observed with several specific agents should be carefully monitored in future research as a potential safety signal.

## Introduction

Anxiety, insomnia, and mood disorders are common during pregnancy [1,2], and benzodiazepines are frequently prescribed to pregnant women to manage these conditions [3]. The worldwide prevalence of benzodiazepine use during pregnancy is approximately 2% [4], and more than 1% of pregnant women in South Korea are prescribed these agents during the first trimester (S1 Fig). Despite their regular use, over the last decades, with a parallel growing focus on newer antidepressants or antipsychotics, benzodiazepines have received limited consideration, resulting in a lack of evidence on the safety of their use [5].

Benzodiazepines readily cross the human placenta and may accumulate in fetal tissues at concentrations higher than those detected in maternal serum [6,7]. The teratogenicity of benzodiazepines is biologically plausible, as they bind to receptors in peripheral tissues as well as the brain and are involved in cell proliferation and differentiation [8]. Given their potential to harm the fetus, regulatory agencies recommend that benzodiazepines should be avoided during pregnancy [9,10]. However, evidence indicating the teratogenicity of benzodiazepines is uncertain as pregnant women are usually excluded from clinical trials. Although different meta-analyses of epidemiological studies have reported no association between benzodiazepines and congenital malformations [11,12], a majority of these studies had significant methodological limitations, including low statistical power due to a small number of exposed women; potential recall and selection bias, as most previous studies were case–control designs; no detailed information regarding benzodiazepine prescriptions (e.g., dosage and indication); and lacked control for important confounders (e.g., psychiatric comorbidities, concomitant medications) [11–20]. Moreover, although pharmacokinetic and pharmacodynamic profiles substantially differ among individual benzodiazepines [21], most studies have interpreted them simply as a class effect, and few studies have demonstrated a dose–response relationship.

Accordingly, to address these limitations and encourage optimal therapeutic decisions for pregnant women, additional research is needed in an adequately large-scale pregnancy cohort. Thus, we conducted a nationwide cohort study in South Korea to examine the association between maternal exposure to benzodiazepines during the first trimester and the risk of major congenital malformations in their offspring.

## Methods

### Data source and study cohort

We conducted a nationwide retrospective cohort study using healthcare data retrieved from the Health Insurance Review and Assessment Service (HIRA) database, which covers 50 million people (approximately 99% of the South Korean population), from 1 July 2009 to 31 December 2019. These data comprise individual-level demographics and all records of diagnosis and healthcare utilization (e.g., drug prescription and medical procedure), provided through inpatient, outpatient, and emergency department visits. In a validation study comparing our database and electronic medical records, the overall positive predictive value of diagnosis records was 82% [22].

Our cohort included all pregnancies resulting in live births from 1 January 2011 to 31 December 2018, identified with procedure codes of delivery (S1 Table). We included all live-born infants who were linked with their mothers and restricted the pregnancy cohort to women aged 20 to 45 years at delivery. We excluded pregnancies diagnosed with a chromosomal abnormality; those with exposure to known teratogenic drugs during the first trimester; and those unexposed to benzodiazepines during the first trimester, but exposed at least once during the 3 months preceding the last menstrual period, to avoid contaminating the unexposed group with women who could have taken benzodiazepines post-pregnancy (S2 Table). Infants were followed from birth to 8 years, death, or the end of the study period (December 2019), whichever came first. We calculated the date of the last menstrual period using an algorithm to estimate the gestational age in administrative databases [23].

The need for informed consent was waived, as this study was conducted using anonymized claims data. This study was approved by the Institutional Review Board of Sungkyunkwan University, South Korea (No. 2021-04-005). We registered the study protocol on ClinicalTrials.gov (NCT04856436).

## Exposure

The exposed group was composed of women who filled at least one benzodiazepine prescription during the first trimester (first 90 days of pregnancy), known as the etiologically relevant period for congenital malformations. The unexposed group comprised women who were not prescribed any benzodiazepine from 3 months before the last menstrual period to the end of the first trimester. Furthermore, we evaluated short-acting (half-life ≤24 h) and long-acting (half-life >24 h) benzodiazepines based on the duration of action [24], as well as the individual medications. To assess a dose–response relationship, we calculated all benzodiazepine doses by converting them to lorazepam-equivalent doses [25,26], and we classified them into 3 mean daily dose groups: <1 mg/day, 1 to 2.5 mg/day, and >2.5 mg/day.

## Outcomes

Major congenital malformations were identified by diagnostic records, according to the ICD-10 codes defined by the European Surveillance of Congenital Anomalies classification (S1 Table) [27]. Major congenital malformations were further categorized into 12 types of organ-specific malformations: (1) nervous system; (2) eye; (3) ear, face, and neck; (4) heart; (5) respiratory system; (6) oral cleft; (7) digestive system; (8) abdominal wall; (9) urinary system; (10) genital organs; (11) limb; and (12) other malformations. The primary outcomes of interest were overall major congenital malformations and heart defects (accounting for most malformations); other types of organ-specific malformations were considered secondary outcomes, owing to the anticipated small number of events. For exploratory analyses, we evaluated the risks of individual categories of heart defects and digestive system, as previous studies have reported that these specific malformations may be correlated with benzodiazepines [15,17–19].

## Covariates

We considered a broad range of covariates as potential confounders or proxies of potential confounders: maternal demographics status (e.g., age and the type of insurance), psychiatric conditions (e.g., bipolar disorder, depression/mood disorder, anxiety, and sleep disorder), maternal conditions (e.g., epilepsy/seizures, headache/migraine, diabetes, hypertension), obstetric conditions (parity, plurality), concomitant medications (e.g., antidepressants, anticonvulsants, antipsychotics, and opioid analgesics), and measures of healthcare utilization (e.g., obstetric comorbidity index [28,29], number of distinct diagnoses, and hospital

admission) (S1 Table). Maternal comorbidities and concomitant medication use were measured from 6 months before the last menstrual period to the end of the first trimester. Measures of healthcare utilization were measured during the 6 months before, but not during, pregnancy to avoid these variables being affected by early awareness of pregnancy, except for obstetric comorbidity index [28,29].

## Statistical analysis

Baseline characteristics of women exposed and those unexposed to benzodiazepines were compared using absolute standardized differences (aSDs; $\geq$0.1 indicates a significant imbalance between the 2 groups). The absolute risks (per 1,000 pregnancies), risk differences, and unadjusted relative risks (RRs) with 95% confidence intervals (CIs) were calculated for each outcome, stratified by exposure to benzodiazepines. Moreover, we calculated the population-attributable fractions (PAFs) to estimate the impact on an outcome of exposure in the study population [30]. In this study, PAF means the fractions of overall or individual congenital malformations in pregnant women attributable to benzodiazepines. We calculated PAF as (O-E)/O, where O is the observed number of outcomes and E is the expected number of outcomes under no exposure [31]. We used the propensity score (PS) fine stratification method to control for potential confounders [32]. Accordingly, the PS for exposure to benzodiazepines versus nonexposure was derived using a logistic regression model, which included all covariates without additional selection. After trimming the observations in nonoverlapping regions of the PS distribution, the exposed women were divided into 50 equally sized strata based on the PS distribution; then, unexposed women were weighted using the distribution of the exposed women for each stratum. We estimated the adjusted RR with 95% CI using a generalized linear model (log-binomial model).

We conducted 7 prespecified subgroup analyses to determine whether the risk of congenital malformations varied across exposure and maternal characteristics. Stratified analyses were performed according to the duration of action, individual benzodiazepines, and mean daily dose group with the 3 categories. Furthermore, we conducted stratified analyses by maternal age at delivery ($\leq$35 and >35 years), multifetal pregnancy, history of epilepsy, and concomitant use of antidepressants.

Additionally, we performed diverse sensitivity analyses for all primary and secondary outcomes to evaluate the robustness of the main findings. First, we redefined the use of exposure as having filled at least 2 benzodiazepine prescriptions during the first trimester. Second, we redefined the outcome definition as the presence of $\geq$2 diagnoses of congenital malformations. Third, we restricted the study cohort to those who had underlying comorbidities related to the indication for benzodiazepines (bipolar disorder, depression/mood disorder, anxiety, sleep disorder, and gastrointestinal disease) to mitigate confounding by indication. Fourth, we restricted the cohort to nulliparous women to account for intraindividual correlations that might arise from repeated measurements of the same women. Fifth, we conducted a negative control analysis by comparing negative exposure control (defined as pregnancies exposed to benzodiazepines between 180 days and 90 days before the last menstrual period, which is not an etiologically relevant window for congenital malformations) with the reference group in the main analysis (pregnancies not exposed to benzodiazepines in the first trimester). If the main finding is subject to residual confounding, we can expect a non-null result from the negative control analysis. Sixth, for outcomes that presented an increased risk, we used a rule-out approach to explore the impact of unmeasured confounders (e.g., maternal smoking status) (S1 Appendix). Lastly, we conducted a quantitative bias analysis based on the probabilistic method to address the impact of selection bias (S2 Appendix). All analyses were conducted

according to the prespecified analysis plan (S1 Protocol). Statistical analyses were performed using SAS version 9.4 (SAS Institute, Cary, NC, USA).

## Results

Our cohort consisted of 3,094,227 pregnancies, of which 40,846 (1.3%) were prescribed at least one benzodiazepine during the first trimester (mean [standard deviation (SD)] age, 32.4 [4.1] years) (Table 1). The benzodiazepine-exposed group had higher medical conditions and concomitant medication use for both psychiatric-related and unrelated conditions (e.g., 12.6%

**Table 1. Cohort characteristics of pregnancies with and without benzodiazepine exposure during the first trimester.**

| Characteristics | Unadjusted | | | PS adjusted* | | |
|---|---|---|---|---|---|---|
| | Benzodiazepines (*n* = 40,846) | Unexposed (*n* = 3,053,381) | aSD† | Benzodiazepines (*n* = 40,844) | Unexposed (*n* = 3,053,207) | aSD† |
| Maternal age, mean (SD), y | 32.4 (4.1) | 32.1 (4.6) | 0.07 | - | - | |
| Maternal age group, n (%) | | | | | | |
| 20–25 | 3,251 (8.0) | 182,928 (6.0) | 0.08 | 3,251 (8.0) | 255,299 (8.4) | 0.01 |
| 26–30 | 10,170 (24.9) | 836,449 (27.4) | 0.06 | 10,170 (24.9) | 750,537 (24.6) | 0.01 |
| 31–35 | 17,218 (42.2) | 1,450,643 (47.5) | 0.11 | 17,216 (42.2) | 1,264,206 (41.4) | 0.02 |
| 36–40 | 8,672 (21.2) | 522,330 (17.1) | 0.10 | 8,672 (21.2) | 662,006 (21.7) | 0.01 |
| 41–45 | 1,535 (3.8) | 61,031 (2.0) | 0.11 | 1,535 (3.8) | 121,159 (4.0) | 0.01 |
| Medical aid recipients, n (%) | 346 (0.8) | 6,569 (0.2) | 0.09 | 346 (0.8) | 24,311 (0.8) | 0.01 |
| Psychiatric conditions, n (%) | | | | | | |
| Bipolar disorder | 657 (1.6) | 2,066 (0.1) | 0.17 | 657 (1.6) | 34,940 (1.1) | 0.04 |
| Depression/mood disorder | 5,121 (12.5) | 23,176 (0.8) | 0.49 | 5,120 (12.5) | 329,620 (10.8) | 0.05 |
| Anxiety | 5,133 (12.6) | 20,301 (0.7) | 0.49 | 5,132 (12.6) | 333,795 (10.9) | 0.05 |
| Sleep disorder | 4,058 (9.9) | 19,277 (0.6) | 0.43 | 4,057 (9.9) | 256,517 (8.4) | 0.05 |
| Nonaffective psychosis | 535 (1.3) | 1,755 (0.1) | 0.15 | 535 (1.3) | 30,776 (1.0) | 0.03 |
| Stress-related disorder | 3,250 (8.0) | 18,658 (0.6) | 0.37 | 3,249 (8.0) | 229,705 (7.5) | 0.02 |
| Eating disorder | 124 (0.3) | 784 (0.0) | 0.07 | 123 (0.3) | 8,097 (0.3) | 0.01 |
| Personality disorder | 87 (0.2) | 374 (0.0) | 0.06 | 87 (0.2) | 5,114 (0.2) | 0.01 |
| Maternal conditions, n (%) | | | | | | |
| Epilepsy/seizure | 505 (1.2) | 4,510 (0.1) | 0.13 | 505 (1.2) | 35,693 (1.2) | 0.01 |
| Headache/migraine | 6,476 (15.9) | 141,294 (4.6) | 0.38 | 6,476 (15.9) | 541,466 (17.7) | 0.05 |
| Diabetes | 553 (1.4) | 18,722 (0.6) | 0.08 | 553 (1.4) | 43,391 (1.4) | 0.01 |
| Hypertension | 694 (1.7) | 21,632 (0.7) | 0.09 | 693 (1.7) | 53,954 (1.8) | 0.01 |
| Renal disease | 207 (0.5) | 8,557 (0.3) | 0.04 | 207 (0.5) | 16,560 (0.5) | 0.00 |
| Gastrointestinal disease | 3,519 (8.6) | 29,324 (1.0) | 0.36 | 3,518 (8.6) | 282,927 (9.3) | 0.02 |
| Alcohol or drug dependence | 341 (0.8) | 1,520 (0.0) | 0.12 | 341 (0.8) | 19,267 (0.6) | 0.02 |
| Tobacco dependence | 2 (0.0) | 36 (0.0) | 0.01 | 2 (0.0) | 174 (0.0) | 0.00 |
| Obstetric conditions, n (%) | | | | | | |
| Nulliparous | 20,477 (50.1) | 1,601,759 (52.5) | 0.10 | 20,476 (50.1) | 1,511,234 (49.5) | 0.01 |
| Multifetal pregnancy | 1,438 (3.5) | 52,903 (1.7) | 0.11 | 1,438 (3.5) | 123,097 (4.0) | 0.03 |
| Concomitant medications, n (%) | | | | | | |
| Antidepressants | 6,910 (16.9) | 43,245 (1.4) | 0.56 | 6,909 (16.9) | 470,554 (15.4) | 0.04 |
| Anxiolytics | 2,744 (6.7) | 98,944 (3.2) | 0.16 | 2,743 (6.7) | 207,339 (6.8) | 0.00 |
| Hypnotics | 3,129 (7.7) | 19,387 (0.6) | 0.36 | 3,128 (7.7) | 198,914 (6.5) | 0.04 |
| Barbiturates | 967 (2.4) | 41,455 (1.4) | 0.07 | 967 (2.4) | 79,843 (2.6) | 0.02 |
| Anticonvulsants | 1,508 (3.7) | 21,158 (0.7) | 0.21 | 1,508 (3.7) | 107,435 (3.5) | 0.01 |
| Antipsychotics | 2,126 (5.2) | 5,842 (0.2) | 0.31 | 2,125 (5.2) | 113,254 (3.7) | 0.07 |

(*Continued*)

**Table 1.** (Continued)

| Characteristics | Unadjusted | | | PS adjusted* | | |
|---|---|---|---|---|---|---|
| | Benzodiazepines (*n* = 40,846) | Unexposed (*n* = 3,053,381) | aSD† | Benzodiazepines (*n* = 40,844) | Unexposed (*n* = 3,053,207) | aSD† |
| Stimulants | 112 (0.3) | 411 (0.0) | 0.07 | 112 (0.3) | 6,767 (0.2) | 0.01 |
| Opioid analgesics | 17,486 (42.8) | 602,813 (19.7) | 0.51 | 17,485 (42.8) | 1,428,633 (46.8) | 0.08 |
| Noninsulin antidiabetic agents | 334 (0.8) | 10,448 (0.3) | 0.06 | 334 (0.8) | 26,611 (0.9) | 0.01 |
| Insulins | 262 (0.6) | 9,811 (0.3) | 0.05 | 262 (0.6) | 20,127 (0.7) | 0.00 |
| Antihypertensives | 4,097 (10.0) | 44,679 (1.5) | 0.37 | 4,096 (10.0) | 287,528 (9.4) | 0.02 |
| Nonsteroidal anti-inflammatory drugs | 26,474 (64.8) | 1,269,496 (41.6) | 0.48 | 26,473 (64.8) | 2,106,157 (69.0) | 0.09 |
| Triptans | 445 (1.1) | 7,030 (0.2) | 0.11 | 445 (1.1) | 35,619 (1.2) | 0.01 |
| Lipid lowering drug | 414 (1.0) | 8,629 (0.3) | 0.09 | 414 (1.0) | 31,005 (1.0) | 0.00 |
| Antithyroid drugs | 360 (0.9) | 17,774 (0.6) | 0.04 | 360 (0.9) | 28,429 (0.9) | 0.01 |
| Thyroid hormones | 1,594 (3.9) | 112,391 (3.7) | 0.01 | 1,594 (3.9) | 121,123 (4.0) | 0.00 |
| Systemic corticosteroids | 10,386 (25.4) | 466,690 (15.3) | 0.25 | 10,386 (25.4) | 828,069 (27.1) | 0.04 |
| Azoles | 12,161 (29.8) | 732,996 (24.0) | 0.13 | 12,160 (29.8) | 928,981 (30.4) | 0.01 |
| Fertility drugs | 2,774 (6.8) | 98,401 (3.2) | 0.16 | 2,774 (6.8) | 237,642 (7.8) | 0.04 |
| **Healthcare utilization** | | | | | | |
| Obstetric comorbidity index, mean (SD) | 0.8 (1.0) | 0.5 (0.8) | 0.26 | 0.8 (1.0) | 0.8 (1.1) | 0.03 |
| Number of distinct diagnoses, mean (SD) | 4.6 (3.3) | 3.1 (2.5) | 0.54 | 4.6 (3.3) | 4.8 (3.2) | 0.04 |
| Number of distinct prescription drugs, excluding benzodiazepines, mean (SD) | 16.4 (12.0) | 10.1 (8.9) | 0.60 | 16.4 (12.0) | 17.0 (11.8) | 0.05 |
| History of emergency room visits, n (%) | 4,165 (10.2) | 174,098 (5.7) | 0.17 | 4,164 (10.2) | 314,654 (10.3) | 0.00 |
| Patients hospitalized, n (%) | 3,473 (8.5) | 155,802 (5.1) | 0.15 | 3,472 (8.5) | 265,536 (8.7) | 0.01 |
| Number of outpatient visits, mean (SD) | 9.5 (9.3) | 5.3 (5.5) | 0.55 | 9.5 (9.3) | 9.6 (9.1) | 0.01 |

aSD, absolute standardized difference; PS, propensity score; SD, standard deviation.

*To account for PS, unexposed observations were weighted using the distribution of the exposed observations among the 50 PS strata. Observations from nonoverlapping regions of the PS distributions were trimmed.

†The value >0.10 indicates a significant imbalance between the exposed and unexposed groups.

exposed versus 0.7% unexposed had an anxiety disorder; 16.9% exposed versus 1.4% unexposed received an antidepressant) than the unexposed group. All cohort characteristics were well balanced between the exposed and unexposed groups after PS adjustment, with an aSD <0.1.

The absolute risk difference for overall malformations was 13.9 per 1,000 pregnancies (65.3 versus 51.4 per 1,000 in the exposed and unexposed groups, respectively) and that for heart defects was 11.5 per 1,000 (38.9 versus 27.4 per 1,000) (Fig 1). The PAFs of overall malformations and heart defects were 0.36% and 0.55%, respectively. The unadjusted RRs increased for overall malformations, heart defects, digestive system defects, abdominal wall defects, urinary system defects, genital defects, and other malformations. After adjustment for potential confounders, the RR estimates shifted substantially toward a null value; however, the risk for overall malformations and heart defects, although small, remained significantly elevated (adjusted RR 1.09 [95% CI 1.05 to 1.13, *p* < 0.001] and 1.15 [1.10 to 1.21, *p* < 0.001], respectively). Of individual heart defects, significant associations were found in cardiac septal defects (adjusted RR 1.13, 95% CI 1.07 to 1.20, *p* < 0.001) and defects of the great arteries (1.28, 95% CI 1.16 to 1.42, *p* < 0.001) (Fig 2).

We observed that the risks for overall malformations were comparable between short- and long-acting benzodiazepines (adjusted RR 1.09 [95% CI 1.03 to 1.14, *p* < 0.001] versus 1.07

| Outcomes | Benzodiazepines (n = 40,846) | | Unexposed (n = 3,053,381) | | $RD_{1000}$* | PAF | Relative risk (95% CI) | | PS-adjusted relative risk (95% CI) | p-value |
|---|---|---|---|---|---|---|---|---|---|---|
| | No. of Events | Risk/1,000 Births | No. of Events | Risk/1,000 Births | | | Unadjusted | PS-adjusted | | |
| **Overall congenital malformations** | 2,667 | 65.3 | 156,896 | 51.4 | 13.9 | 0.36% | 1.27 (1.22–1.32) | 1.09 (1.05–1.13) | | <0.001 |
| Nervous system | 155 | 3.8 | 10,032 | 3.3 | 0.5 | 0.20% | 1.15 (0.99–1.35) | 1.05 (0.89–1.24) | | 0.571 |
| Eye | 57 | 1.4 | 4,044 | 1.3 | 0.1 | 0.07% | 1.05 (0.81–1.37) | 0.99 (0.75–1.29) | | 0.921 |
| Ear, face, and neck | 27 | 0.7 | 1,835 | 0.6 | 0.1 | 0.13% | 1.10 (0.75–1.61) | 1.04 (0.70–1.53) | | 0.846 |
| Heart | 1,588 | 38.9 | 83,584 | 27.4 | 11.5 | 0.55% | 1.42 (1.35–1.49) | 1.15 (1.10–1.21) | | <0.001 |
| Respiratory system | 23 | 0.6 | 1,658 | 0.5 | 0.0 | 0.05% | 1.04 (0.69–1.56) | 0.91 (0.59–1.40) | | 0.679 |
| Oral clefts | 60 | 1.5 | 4,252 | 1.4 | 0.1 | 0.07% | 1.05 (0.82–1.36) | 0.89 (0.68–1.16) | | 0.394 |
| Digestive system | 142 | 3.5 | 8,697 | 2.8 | 0.6 | 0.29% | 1.22 (1.03–1.44) | 1.13 (0.95–1.35) | | 0.156 |
| Abdominal wall | 27 | 0.7 | 1,284 | 0.4 | 0.2 | 0.75% | 1.57 (1.07–2.30) | 1.08 (0.72–1.63) | | 0.701 |
| Urinary system | 258 | 6.3 | 16,876 | 5.5 | 0.8 | 0.19% | 1.14 (1.01–1.29) | 1.02 (0.90–1.16) | | 0.752 |
| Genital organs | 227 | 5.6 | 13,514 | 4.4 | 1.1 | 0.34% | 1.26 (1.10–1.43) | 1.11 (0.97–1.27) | | 0.130 |
| Limb | 184 | 4.5 | 12,956 | 4.2 | 0.3 | 0.08% | 1.06 (0.92–1.23) | 0.94 (0.80–1.09) | | 0.385 |
| Others | 149 | 3.6 | 8,927 | 2.9 | 0.7 | 0.33% | 1.25 (1.06–1.47) | 1.13 (0.95–1.34) | | 0.163 |

0.5   1.0   2.0

**Fig 1. Absolute and relative risks of congenital malformations in infants following maternal exposure to benzodiazepines during the first trimester.** CI, confidence interval; PAF, population attributable fraction; PS, propensity score; *$RD_{1000}$, risk difference per 1,000 births.

[95% CI 1.01 to 1.13, p = 0.014]) (Figs 3, S2 and S3). The risk of overall malformations, although small but significantly, increased with midazolam, diazepam, flunitrazepam, and chlordiazepoxide administration; the risk of heart defects increased with midazolam, alprazolam, diazepam, flunitrazepam, and ethyl loflazepate administration (range of adjusted RRs for the primary outcomes: 1.08 to 2.43). The increased risk for overall malformations and heart defects was highest in the group of a mean daily lorazepam-equivalent dose of >2.5 mg/day (adjusted RR 1.26 [95% CI 1.17 to 1.36, p < 0.001] and 1.31 [95% CI 1.19 to 1.45, p < 0.001],

| Outcomes | Benzodiazepines (n = 40,846) | | Unexposed (n = 3,053,381) | | $RD_{1000}$* | PAF | Relative risk (95% CI) | | PS-adjusted relative risk (95% CI) | p-value |
|---|---|---|---|---|---|---|---|---|---|---|
| | No. of Events | Risk /1,000 Births | No. of Events | Risk /1,000 Births | | | Unadjusted | PS-adjusted | | |
| **Subgroups of congenital heart defects** | | | | | | | | | | |
| Cardiac chambers and connections | 21 | 0.5 | 1,638 | 0.5 | 0.0 | -0.05% | 0.96 (0.62–1.47) | 0.76 (0.49–1.20) | | 0.238 |
| Cardiac septum | 1,244 | 30.5 | 66,967 | 21.9 | 8.5 | 0.51% | 1.39 (1.31–1.47) | 1.13 (1.07–1.20) | | <0.001 |
| Pulmonary and tricuspid valve | 67 | 1.6 | 3,459 | 1.1 | 0.5 | 0.59% | 1.45 (1.14–1.84) | 1.24 (0.96–1.60) | | 0.094 |
| Aortic and mitral valve | 18 | 0.4 | 1,109 | 0.4 | 0.1 | 0.28% | 1.21 (0.76–1.93) | 0.97 (0.60–1.58) | | 0.905 |
| Other heart defects | 96 | 2.4 | 6,509 | 2.1 | 0.2 | 0.14% | 1.10 (0.90–1.35) | 0.91 (0.74–1.12) | | 0.366 |
| Great arteries | 411 | 10.1 | 18,704 | 6.1 | 3.9 | 0.84% | 1.64 (1.49–1.81) | 1.28 (1.16–1.42) | | <0.001 |
| Great veins | 12 | 0.3 | 677 | 0.2 | 0.1 | 0.43% | 1.33 (0.75–2.34) | 1.17 (0.65–2.13) | | 0.596 |
| **Subgroups of digestive system defects** | | | | | | | | | | |
| Tongue, mouth, and pharynx | 28 | 0.7 | 2,579 | 0.8 | -0.2 | -0.25% | 0.81 (0.56–1.18) | 0.80 (0.55–1.18) | | 0.265 |
| Oesophagus | 11 | 0.3 | 488 | 0.2 | 0.1 | 0.90% | 1.69 (0.93–3.06) | 1.56 (0.84–2.89) | | 0.159 |
| Upper alimentary tract | 1 | 0.0 | 135 | 0.0 | 0.0 | -0.59% | 0.55 (0.08–3.96) | 0.49 (0.07–3.67) | | 0.490 |
| Small intestine | 17 | 0.4 | 847 | 0.3 | 0.1 | 0.66% | 1.50 (0.93–2.42) | 1.22 (0.74–2.03) | | 0.433 |
| Large intestine | 26 | 0.6 | 1,313 | 0.4 | 0.2 | 0.63% | 1.48 (1.00–2.18) | 1.34 (0.89–2.03) | | 0.159 |
| Other defects of intestine | 48 | 1.2 | 2,625 | 0.9 | 0.3 | 0.48% | 1.37 (1.03–1.82) | 1.22 (0.91–1.65) | | 0.186 |
| Gallbladder, bile ducts and liver | 19 | 0.5 | 1,009 | 0.3 | 0.1 | 0.54% | 1.41 (0.89–2.22) | 1.33 (0.83–2.13) | | 0.229 |
| Other defects of digestive system | 1 | 0.0 | 149 | 0.0 | 0.0 | -0.66% | 0.50 (0.07–3.59) | 0.51 (0.07–3.70) | | 0.507 |
| Diaphragmatic hernia | 10 | 0.2 | 504 | 0.2 | 0.1 | 0.63% | 1.48 (0.79–2.77) | 1.36 (0.71–2.60) | | 0.356 |

0.3   0.5   1.0   2.0

**Fig 2. Risks of individual categories of heart and digestive system defects in infants following maternal exposure to benzodiazepines during the first trimester.** CI, confidence interval; PAF, population attributable fraction; PS, propensity score; *$RD_{1000}$, risk difference per 1,000 births.

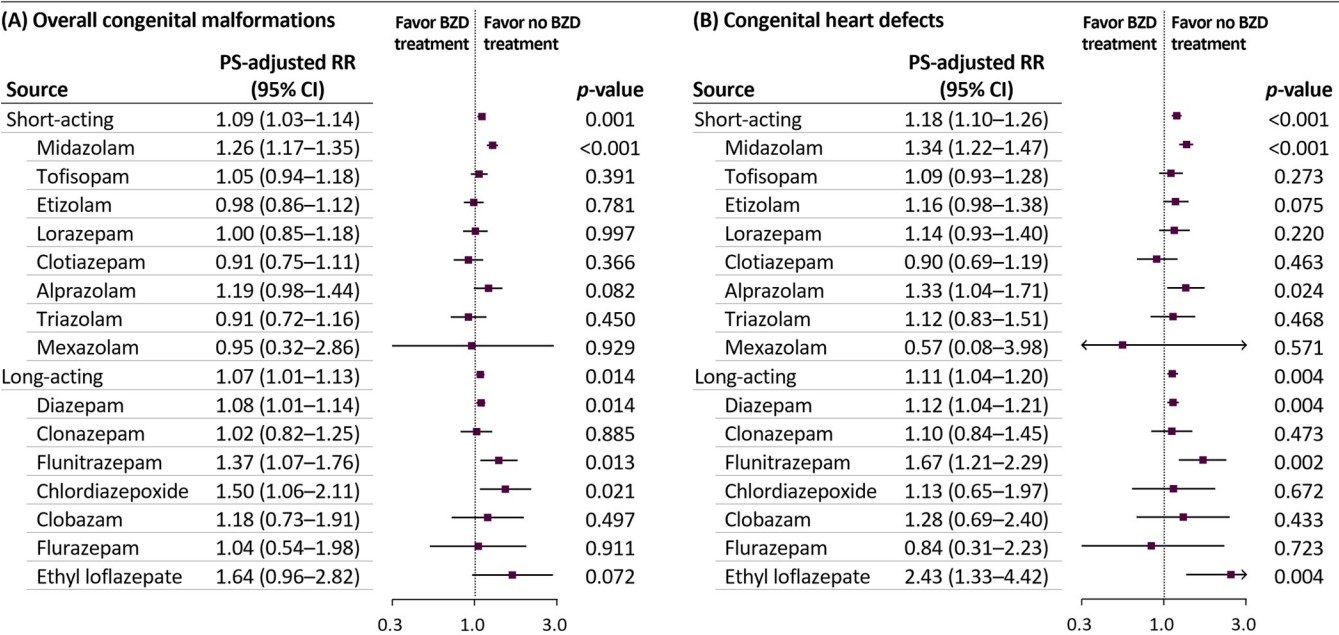

**Fig 3. Risks of congenital malformations in infants according to individual benzodiazepine exposure during the first trimester.** BZD, benzodiazepine; CI, confidence interval; RR, relative risk; PS, propensity score. Clorazepate was not analyzed owing to the small sample size (*n* = 14).

respectively) (Fig 4). In addition, the RR for both primary outcomes were higher among women aged >35 years, those with multifetal pregnancy, and those without a history of epilepsy than in their counterparts. The RR for heart defects was higher among women who used both benzodiazepines and antidepressants than in women unexposed to these agents.

Our main findings remained largely consistent in all sensitivity analyses when redefining exposure and outcome and restricting to pregnancies with benzodiazepine-related underlying diseases and nulliparous pregnancies, as well as the negative control analysis (Figs 5, S4, S5 and S6).

## Discussion

### Main findings

In this nationwide cohort study of approximately 3.1 million pregnancies, first-trimester benzodiazepine use was associated with a small increased risk of overall malformations, particularly heart defects. The risk of primary outcomes increased with a higher mean daily dose of benzodiazepines (>2.5 mg/day of lorazepam-equivalent dose), suggesting a dose–response relationship. Although these risks were similar between short- and long-acting benzodiazepines, a small but significant increase in risk was detected with several specific agents. For the other 11 organ-specific malformation types, we found no significant increase in the risk associated with first-trimester benzodiazepine exposure. Based on the upper limit of the 95% CI from adjusted estimates, the maximum observed risk was 63% for the abdominal wall defects (RR 1.08, 95% CI 0.72 to 1.63); thus, we could rule out the possibility of a more than 63% increase in the risk for all organ-specific malformations. Our findings were consistent across diverse sensitivity analyses, and the null finding in the negative control analysis strengthened the suggestion that the estimate is unlikely to be due to residual confounding.

A recent study has reported an increased risk of spontaneous abortion associated with benzodiazepine exposure in early pregnancy (adjusted odds ratio 1.85; 95% CI 1.61 to 2.12),

| Subgroups | Benzodiazepine | | Unexposed | | Relative risk (95% CI) | | PS-adjusted relative risk (95% CI) | p-value |
|---|---|---|---|---|---|---|---|---|
| | No. of Events | No. of Births | No. of Events | No. of Births | Unadjusted | PS-adjusted | | |
| **Overall congenital malformations** | | | | | | | | |
| Total | 2,667 | 40,846 | 156,896 | 3,053,381 | 1.27 (1.22–1.32) | 1.09 (1.05–1.13) | | <0.001 |
| **Dose analysis*** | | | | | | | | |
| <1 mg/day | 1,154 | 19,668 | 156,896 | 3,053,381 | 1.14 (1.08–1.21) | 1.05 (0.99–1.12) | | 0.077 |
| 1–2.5 mg/day | 856 | 13,079 | 156,896 | 3,053,381 | 1.27 (1.19–1.36) | 1.04 (0.98–1.12) | | 0.215 |
| >2.5 mg/day | 657 | 8,099 | 156,896 | 3,053,381 | 1.58 (1.47–1.70) | 1.26 (1.17–1.36) | | <0.001 |
| **Maternal age** | | | | | | | | |
| ≤35 years | 1,856 | 30,639 | 122,854 | 2,470,020 | 1.22 (1.16–1.27) | 1.07 (1.02–1.12) | | 0.003 |
| >35 years | 811 | 10,207 | 34,042 | 583,361 | 1.36 (1.27–1.46) | 1.13 (1.05–1.22) | | 0.001 |
| **Multifetal pregnancy** | | | | | | | | |
| No | 2,354 | 39,408 | 148,354 | 3,000,478 | 1.21 (1.16–1.26) | 1.08 (1.04–1.13) | | <0.001 |
| Yes | 313 | 1,438 | 8,542 | 52,903 | 1.35 (1.22–1.49) | 1.18 (1.05–1.31) | | 0.004 |
| **Epilepsy** | | | | | | | | |
| No | 2,626 | 40,341 | 156,536 | 3,048,871 | 1.27 (1.22–1.32) | 1.09 (1.05–1.13) | | <0.001 |
| Yes | 41 | 505 | 360 | 4,510 | 1.02 (0.75–1.39) | 1.01 (0.70–1.46) | | 0.968 |
| **Combination of antidepressants** | | | | | | | | |
| No | 2,322 | 35,385 | 156,896 | 3,053,381 | 1.28 (1.23–1.33) | 1.11 (1.06–1.15) | | <0.001 |
| Yes | 345 | 5,461 | 156,896 | 3,053,381 | 1.23 (1.11–1.36) | 1.04 (0.93–1.18) | | 0.490 |
| **Congenital heart defects** | | | | | | | | |
| Total | 1,588 | 40,846 | 83,584 | 3,053,381 | 1.42 (1.35–1.49) | 1.15 (1.10–1.21) | | <0.001 |
| **Dose analysis*** | | | | | | | | |
| <1 mg/day | 668 | 19,668 | 156,896 | 3,053,381 | 1.24 (1.15–1.34) | 1.12 (1.04–1.21) | | 0.004 |
| 1–2.5 mg/day | 526 | 13,079 | 156,896 | 3,053,381 | 1.47 (1.35–1.60) | 1.13 (1.03–1.23) | | 0.008 |
| >2.5 mg/day | 394 | 8,099 | 156,896 | 3,053,381 | 1.78 (1.61–1.96) | 1.31 (1.19–1.45) | | <0.001 |
| **Maternal age** | | | | | | | | |
| ≤35 years | 1,073 | 30,639 | 64,383 | 2,470,020 | 1.34 (1.27–1.43) | 1.14 (1.07–1.21) | | <0.001 |
| >35 years | 515 | 10,207 | 19,201 | 583,361 | 1.53 (1.41–1.67) | 1.20 (1.10–1.32) | | <0.001 |
| **Multifetal pregnancy** | | | | | | | | |
| No | 1339 | 39,408 | 77,747 | 3,000,478 | 1.31 (1.24–1.38) | 1.14 (1.07–1.20) | | <0.001 |
| Yes | 249 | 1,438 | 5,837 | 52,903 | 1.57 (1.40–1.76) | 1.31 (1.15–1.49) | | <0.001 |
| **Epilepsy** | | | | | | | | |
| No | 1,562 | 40,341 | 83,378 | 3,048,871 | 1.42 (1.35–1.49) | 1.16 (1.10–1.22) | | <0.001 |
| Yes | 26 | 505 | 206 | 4,501 | 1.13 (0.76–1.68) | 1.02 (0.62–1.67) | | 0.945 |
| **Combination of antidepressants** | | | | | | | | |
| No | 1,358 | 35,385 | 83,584 | 3,053,381 | 1.40 (1.33–1.48) | 1.16 (1.10–1.23) | | <0.001 |
| Yes | 230 | 5,461 | 83,584 | 3,053,381 | 1.54 (1.36–1.75) | 1.26 (1.08–1.46) | | 0.003 |

0.5    1.0    2.0

**Fig 4. Risks of congenital malformations in infants following maternal exposure to benzodiazepines in the first trimester: subgroup analyses.** CI, confidence interval; PS, propensity score. *Mean daily dose based on the lorazepam-equivalent dose.

indicating the gestational teratogenicity of benzodiazepines [33]. In the latest meta-analysis, which included 8 cohort studies, odds ratios of 1.13 (95% CI 0.99 to 1.30) and 1.27 (95% CI 0.98 to 1.65) for overall malformations and heart defects, respectively, were reported (5,195 exposed pregnancies) [11]; the magnitude of the observed effect was similar to that in our study. Although the meta-analysis concluded that no association existed between benzodiazepine use in pregnancy and congenital malformations with null findings, the lower limit of the 95% CI of estimates was close to 1. Moreover, most previous studies had limited power and were insufficient to adequately assess the potential risk of organ-specific malformations. Notably, our study had a much larger cohort of pregnancies (40,846 exposed pregnancies) than any study published to date, thus expanding on previous findings by providing more precise estimates, as well as controlling numerous potential confounders.

| | Benzodiazepine | | Unexposed | | Relative risk (95% CI) | | PS-adjusted relative risk (95% CI) | p-value |
|---|---|---|---|---|---|---|---|---|
| | No. of Events | No. of Births | No. of Events | No. of Births | Unadjusted | PS-adjusted | | |
| **Overall congenital malformations** | | | | | | | | |
| **Main analysis** | 2,667 | 40,846 | 156,896 | 3,053,381 | 1.27 (1.22–1.32) | 1.09 (1.05–1.13) | | <0.001 |
| **Sensitivity analyses** | | | | | | | | |
| ≥2 prescriptions of exposure | 1,690 | 27,809 | 156,896 | 3,053,381 | 1.18 (1.13–1.24) | 1.06 (1.01–1.12) | | 0.012 |
| ≥2 diagnoses of outcome | 1,643 | 40,846 | 93,602 | 3,053,381 | 1.31 (1.25–1.38) | 1.10 (1.05–1.16) | | <0.001 |
| Restriction to women with main indications | 734 | 11,603 | 4,303 | 74,129 | 1.09 (1.01–1.18) | 1.04 (0.96–1.12) | | 0.375 |
| Restriction to nulliparous women | 1,448 | 20,477 | 83,422 | 1,601,759 | 1.36 (1.29–1.43) | 1.12 (1.06–1.18) | | <0.001 |
| Negative control analysis | 5,951 | 105,912 | 149,447 | 2,922,168 | 1.10 (1.07–1.13) | 1.00 (0.97–1.02) | | 0.829 |
| **Congenital heart defects** | | | | | | | | |
| **Main analysis** | 1,588 | 40,846 | 83,584 | 3,053,381 | 1.42 (1.35–1.49) | 1.15 (1.10–1.21) | | <0.001 |
| **Sensitivity analyses** | | | | | | | | |
| ≥2 prescriptions of exposure | 989 | 27,809 | 83,584 | 3,053,381 | 1.30 (1.22–1.38) | 1.13 (1.06–1.21) | | <0.001 |
| ≥2 diagnoses of outcome | 1,006 | 40,846 | 51,156 | 3,053,381 | 1.47 (1.38–1.56) | 1.19 (1.11–1.27) | | <0.001 |
| Restriction to women with main indications | 451 | 11,603 | 2,378 | 74,129 | 1.21 (1.10–1.34) | 1.15 (1.03–1.27) | | 0.012 |
| Restriction to nulliparous women | 891 | 20,477 | 44,782 | 1,601,759 | 1.56 (1.46–1.66) | 1.21 (1.13–1.30) | | <0.001 |
| Negative control analysis | 3,183 | 105,912 | 79,605 | 2,922,168 | 1.10 (1.07–1.14) | 0.99 (0.96–1.03) | | 0.760 |

0.5    1.0    2.0

**Fig 5. Risks of congenital malformations in infants following maternal exposure to benzodiazepines in the first trimester: sensitivity analyses.** CI, confidence interval; PS, propensity score.

An increased risk of oral cleft, which was reported in early studies evaluating in utero benzodiazepine exposure [34,35], was not observed in our study and has also not been observed in recent studies [15,36]. Three previous studies have suggested an increased risk of digestive system defects associated with benzodiazepines [17–19], but the risk was not confirmed in our study. Although a study using the Swedish birth registry has reported an increased risk of pyloric stenosis (odds ratio 3.31, 95% CI 1.53 to 7.84), the result was based on 8 events among 2,537 infants exposed to benzodiazepines [19]. In the present study, only one case of the upper alimentary tract defect was reported among 40,846 pregnancies exposed to benzodiazepines.

Furthermore, we observed a small but significantly increased risk of congenital malformations in the high-dose group (the mean daily lorazepam-equivalent dose >2.5 mg/day, which is higher than the daily dose defined by the World Health Organization) [37]. According to literature, neonates can slowly metabolize small doses of benzodiazepine; however, the drug persists at pharmacologically active concentrations for at least 1 week when high doses are administered to the mother [16,38]. The benzodiazepine residues that exceed the fetal metabolic capacity might impact the incidence of congenital malformations. Furthermore, we observed increased risks for overall or cardiac malformations with several specific benzodiazepines. The increased risk observed with some specific agents should be prudently interpreted, as no clear biological mechanism can explain these risks and chance finding cannot be excluded. Accordingly, this finding could be construed as a safety signal that should be carefully monitored in future studies. Moreover, the risk of heart defects associated with alprazolam use has been described previously [14,16], reported a nearly doubled risk (odds ratio 2.43, 95% CI 1.42 to 4.15); however, the study failed to consider the indication of use and psychiatric comorbidities, which may result in increased effect size in the exposure versus nonexposure groups [19,39].

## Strengths and limitations

To our knowledge, this is the largest cohort study evaluating the association between first-trimester benzodiazepine exposure and congenital malformations. The large sample size allowed

us to evaluate the risk of rare malformations stratified by individual agents and specific dose groups. Additionally, we used a nationwide database representing the entire population of South Korea, which allowed us to generate generalizable real-world evidence. Our study also had potential limitations. First, misclassification of exposure is possible. Thus, we redefined exposure as at least 2 filled benzodiazepine prescriptions during the first trimester on the assumption that if a woman refilled the prescription, she probably took them. Second, outcome misclassification is possible. Therefore, we conducted a sensitivity analysis requiring at least 2 diagnoses, which increased the likelihood that outcomes reflect the actual occurrence of congenital malformations, and the results were consistent with the main findings. Third, our results could be influenced by unmeasured confounders despite the adjustment for abundant confounders. To address this concern, we conducted a negative control analysis, and this analysis revealed no association, suggesting that our main finding was not attributed to residual confounding. In addition, we used the rule-out approach to explore the effect of unmeasured confounders; the result indicated that it was unlikely that the unmeasured confounder would explain the observed association (S1 Appendix). Fourth, our findings could be affected by confounding by indication, as we used the unexposed group as the reference group. However, the results were consistent with the main findings when restricting the study cohort to women who had underlying disease related to the indication for benzodiazepine. Fifth, our study cohort included live births only, which might lead to selection bias because severe malformations that resulted in pregnancy terminations would be missed. Therefore, we conducted quantitative bias analysis, and the corrected RR for the primary outcomes remained below 1.4, under the most extreme scenario, suggesting that the effect of such selection bias is minimal (S2 Appendix).

## Conclusions

In this nationwide cohort study, benzodiazepine use during the first trimester was associated with a small increased risk of overall malformations and heart defects, particularly in the high daily dose group (at doses higher than the usual daily dose). However, the absolute risks and population attributable fractions were modest. Our findings suggest that the benefits of benzodiazepines for their major indications must be considered despite the potential risks. Nonetheless, to minimize the potential risk, alternative nonpharmacological strategies could be considered for managing anxiety and insomnia during pregnancy; if benzodiazepines are necessary, the lowest effective dosage should be prescribed during early pregnancy.

## Supporting information

**S1 STROBE Checklist. STROBE, Strengthening the Reporting of Observational Studies in Epidemiology.**
(DOC)

**S1 Table. Codes used to define the inclusion/exclusion criteria, exposures, outcomes of interest, maternal comorbidities, and concomitant medications.**
(DOCX)

**S2 Table. Selection of the study cohort using the HIRA database between 1 July 2009 and 31 December 2019.**
(DOCX)

**S1 Fig. Frequency of pregnancies exposed to benzodiazepines during the first trimester in South Korea between 2011 and 2018.**
(DOCX)

**S2 Fig. Absolute and relative risks of overall congenital malformations in infants according to individual benzodiazepine exposure during the first trimester.**
(DOCX)

**S3 Fig. Absolute and relative risks of congenital heart defects in infants according to individual benzodiazepine exposure during the first trimester.**
(DOCX)

**S4 Fig. Risks of congenital malformations in infants following maternal exposure to benzodiazepines during the first trimester: sensitivity analyses.**
(DOCX)

**S5 Fig. Risks of congenital malformations in infants following maternal exposure to benzodiazepines during the first trimester: sensitivity analyses II.**
(DOCX)

**S6 Fig. Risks of congenital malformations in infants following maternal exposure to benzodiazepines during the first trimester: sensitivity analyses III.**
(DOCX)

**S1 Appendix. Sensitivity analysis of unmeasured confounders (rule-out approach).**
(DOCX)

**S2 Appendix. Potential effect of including live births only.**
(DOCX)

**S1 Protocol. Summary protocol.**
(DOCX)

## Author Contributions

**Conceptualization:** Yunha Noh, Jun Soo Kwon, Seung-Ah Choe, Ju-Young Shin.

**Data curation:** Hyesung Lee, Jungmi Chae, Dong-Sook Kim.

**Formal analysis:** Yunha Noh, Hyesung Lee, Ahhyung Choi, Jungmi Chae, Dong-Sook Kim.

**Funding acquisition:** Yunha Noh, Ju-Young Shin.

**Investigation:** Yunha Noh, Hyesung Lee, Ahhyung Choi, Jun Soo Kwon, Seung-Ah Choe, Jungmi Chae, Dong-Sook Kim, Ju-Young Shin.

**Methodology:** Yunha Noh, Hyesung Lee, Ahhyung Choi, Jun Soo Kwon, Seung-Ah Choe, Ju-Young Shin.

**Project administration:** Yunha Noh, Dong-Sook Kim, Ju-Young Shin.

**Resources:** Dong-Sook Kim.

**Supervision:** Dong-Sook Kim, Ju-Young Shin.

**Validation:** Dong-Sook Kim, Ju-Young Shin.

**Visualization:** Yunha Noh, Ahhyung Choi.

**Writing – original draft:** Yunha Noh, Hyesung Lee.

**Writing – review & editing:** Yunha Noh, Hyesung Lee, Ahhyung Choi, Jun Soo Kwon, Seung-Ah Choe, Jungmi Chae, Dong-Sook Kim, Ju-Young Shin.

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
