## [Editor Report · Decision Letter 0]

28 Oct 2021

Dear Dr Shin, 

Thank you for submitting your manuscript entitled "Association Between First-trimester Exposure to Benzodiazepines and Risk of Congenital Malformations in Offspring: A Population-Based Cohort Study" for consideration by PLOS Medicine.

Your manuscript has now been evaluated by the PLOS Medicine editorial staff and I am writing to let you know that we would like to send your submission for external assessment.

However, we first need you to complete your submission by providing the metadata that is required for full assessment. To this end, please login to Editorial Manager where you will find the paper in the 'Submissions Needing Revisions' folder on your homepage. Please click 'Revise Submission' from the Action Links and complete all additional questions in the submission questionnaire.

Please re-submit your manuscript within two working days, i.e. by Nov 01 2021 11:59PM.

Once your full submission is complete, your paper will undergo a series of checks in preparation for assessment. 

Kind regards,

Richard Turner, PhD

rturner@plos.org

---

## [Decision Letter · Decision Letter 1]

23 Dec 2021

Dear Dr. Shin,

Thank you very much for submitting your manuscript "Association Between First-trimester Exposure to Benzodiazepines and Risk of Congenital Malformations in Offspring: A Population-Based Cohort Study" (PMEDICINE-D-21-04507R1) for consideration at PLOS Medicine. 

Your paper was discussed with an academic editor with relevant expertise and sent to independent reviewers, including a statistical reviewer. The reviews are appended at the bottom of this email and any accompanying reviewer attachments can be seen via the link below:

[LINK]

In light of these reviews, we will not be able to accept the manuscript for publication in the journal in its current form, but we would like to invite you to submit a revised version that addresses the reviewers' and editors' comments fully. You will appreciate that we cannot make a decision about publication until we have seen the revised manuscript and your response, and we expect to seek re-review by one or more of the reviewers. 

We hope to receive your revised manuscript by Jan 24 2022 11:59PM. Please email us (plosmedicine@plos.org) if you have any questions or concerns.

Please let me know if you have any questions, and we look forward to receiving your revised manuscript. 

Sincerely,

Richard Turner, PhD

Senior editor, PLOS Medicine

rturner@plos.org

Please remove the information on funding, competing interests and data availability from the title page. This information will appear in the article metadata in the event of publication, via entries in the submission form. 

In the abstract and throughout the text, please include p values alongside 95% CI, where available. 

At line 72, should that be "95% confidence intervals"?

Please quote aggregate demographic details for study participants in the abstract.

Please add a new final sentence to the "Methods and findings" subsection of your abstract, which should begin "Study limitations include ..." or similar, and should quote 2-3 of the study's main limitations. 

At line 86, please begin the sentence "In this study, we found that ..." or similar.

You mention a study protocol in the Methods section. If available, please include the document as an attachment, labelled "S2_Protocol" or similar, referred to in the text. 

Please highlight analyses that were not prespecified. 

Throughout the main text, please reformat the reference call-outs as follows: "... during pregnancy [9,10]. However, evidence ..." (noting the absence of spaces within the square brackets). 

In the reference list, please remove the copyright information from reference 9. 

In reference 15, please make the journal name abbreviation "BMJ"; and for reference 20 use the journal name abbreviation "PLoS ONE"; and "JAMA" for reference 37. 

Thank you for including a completed STROBE checklist. Please rename the attachment "S1_STROBE_Checklist" or similar, and refer to it by this label in the Methods section (main text). 

Comments from academic editor:

This paper addresses a clinically relevant question and the methods seem generally sound. The lack of information on terminations of pregnancy is the major limitation. 

I agree with the reviewer re the comment on inclusion of livebirths only. If there was any opportunity to do an additional analysis with stillbirth/miscarriage/terminations that would strengthen the study (if not possible this should be discussed as a limitation).

An additional small point- could the authors explain rationale for excluding pregnancy in women <20y?

Comments from the reviewers:

*** Reviewer #1: 

Statistical review

This paper reports a retrospective cohort study assessing the association between prescription of benzodiazepine during first trimester and congenital malformations at birth. The results show a small but significant association. Many sensitivity results are provided that suggest the main results are robust.

I have some comments that I have listed below.

1. Abstract: I think it's fair to put that 'is associated' rather than 'may be associated' as long as association is not taken to be the same as causality. Would be good to emphasise the low absolute risk?

2. Methods page 7 - please clarify somewhere (and in the title) that this is a retrospective cohort study.

3. Page 7 - why was the analysis restricted to live births? Is there no data on miscarriage/stillbirth? I see this is investigated in the supplementary material, although only in the direction of the main analysis result being underestimated. If the data is available then it might be informative to include a sensitivity analysis that includes miscarriage/stillbirth in the overall outcome of any abnormality.

4. Statistical methods - was a pre-specified analysis plan written? Propensity score matching has so many different methods to implement that it would be useful to know that the reported results were using the pre-planned method.

5. Page 9/10: How complete were covariates and, if there were non-negligible proportions of missing data, how were PS estimated in the presence of missing covariates?

6. Page 10 - was the log-binomial model adjusted for the covariates or just matched on propensity score?

7. Page 10/11 - I do not follow the negative control analysis, is this comparing women exposed 180-90 days before vs those never exposed at any point? I would rephrase the statement on line 238 to be less definitive, e.g. 'If the main analysis were subject to residual confounding then we would expect the negative control analysis to give similar results'?

8. Results - just as a suggestion, it would be of interest to estimate the proportion of abnormalities that are explained by benzodiazepine (i.e. the population attributable fraction).

James Wason

*** Reviewer #2: 

The manuscript does an interesting job generating scientific evidence on a significant topic on human health. The aim of the manuscript was to evaluate the association between first trimester benzodiazepine use and the risk of major congenital malformations. For that, authors have acceded to Korea's nationwide healthcare database, and conducted a population-based cohort study of women who gave birth during 2011-2018 and their live-born infants.

The manuscript it is well written, its design, its methodology and statistical approach applied on data it is sound according to statistical analysis are applying actually in this field.

General comments:

The main concern it is about how to interpret "congenital malformations" as main outcome. Congenital malformations are a wide range of heterogeneous conditions with highly heterogeneous causes.

It would be very usefull if the manuscript could be presented disaggregating the risks over a reduced group of specific categories of congenital malformations. Probably if authors could focus on a list of specific categories (or groups of categories), such as their are made for heart defects, the manuscript will produce a more precise information. Authors show this kind of results but in supplementary tables. Please consider if it could be posible to select a group of specific categories (or group of) and them present it as main results.

-When authors have defined the exposition, and therefore exposed and unexposed group, have been women in the exposed group exclusively exposed to benzodiazepines? or they could be exposed to another drug also? the same question is valid for the unexposed group, were those women exposed to other drug or none drug?

Authors would agree that theoretically there are four possibilities of exposition:

a-exposed only to Benzodiazepines 

b-exposed to benzodiazepines + other drug 

c-exposed to other/s drug/s

d-none drug exposed during pregnancy

Please add information about how frequent it is the exposition to other drugs in this groups. Since depending on how the relative risk are stablished with any combination with this groups, this RR could be interpreted in different ways. Please fell free to consider the following reference if it is useful: DOI: 10.1371/journal.pone.0046626 

Minor comments or questions:

-Have authors considered to obtain Population attributable fraction as an indicator of benzodiazepines use's impact?

-When have mentioned covariates: have you information related to socioeconomic status, socioeconomic conditions, maternal education or similar? since these are closely related to the occurrence of specific birth defects, such as oral cleft, among others.

-When the relative risks for overall congenital malformations were obtained, were the congenital heart defects excluded?

-Have information on prenatal diagnosis for congenital malformation on these pregnancies? could this be another source of bias? 

*** Reviewer #3: 

This is an interesting manuscript with a purpose to "evaluate the association between first-trimester benzodiazepine use and the risk of major congenital malformations." This was a retrospective cohort study using a national database (the Health Insurance Review and Assessment Service (HIRA) database which covers approximately 99% of the South Korean population). 

1. In the abstract "We determined the relative risks (RRs) and confidential internal (CI) of overall congenital malformations and..." Should this be confidence interval?

2. In the Introduction "The worldwide prevalence of benzodiazepines during pregnancy is approximately 2%,(4) and" Should this be benzodiazepine use?

3. For Reference #22 do the authors have the journal and identifying information on where this reference was published? Do the authors have data on the validity of prescription drug entry into the database? Could the authors supply more information about how the data on prescription drugs is entered into the database and its reliability?

*** Reviewer #4: 

This is a well-designed retrospective cohort study of Korean insurance data that assesses the impact of first-trimester exposure of benzodiazepines via active prescriptions on the risk of congenital malformations. The authors found a statistically significant difference in rate of overall and cardiac anomalies. The analysis for these outcomes is robust, including adjusting for 45 confounders. Though the findings are statistically significant, the increase is nominal and not likely clinically relevant, so I would recommend a tempering of language used with suggestions below. 

Abstract:

1. Line 81: Would say "suggesting" rather than "indicating" as the incremental increase in adjusted RR is modest at best. 

2. Line 81: I would make sure to clarify the degree of significance of the findings such as a "small but significant."

3. Line 87-90: I would go so far as to say that in most cases "the benefit of benzodiazepines is likely to outweigh the nominal increased risk" and that the lowest effective dose should be prescribed to "minimize risk"

Author summary:

1. Line 102-103: Suggest "slightly" as opposed to "particularly" and "suggesting a" instead of "indicating the"

2. Line 104-: Suggest "a small but significant increased risk" 

Methods:

1. Line 148 and 153: Why do the database dates and dates of the cohort differ?

2. Line 156: What is the justification for this age range selection?

3. Line 156-160: Was the possibility of substance use disorder also considered and controlled for? (e.g. not prescription based drug use) 

4. Recommend a statistician review the analyses performed though seem appropriate. 

Results/Discussion: 

1. Line 276/308/337: Again suggest "small but significant(ly)"

2. Line 311-312: I do not understand what this line means or what data this 63% comes from.

***

[LINK]

---

## [Decision Letter · Decision Letter 2]

9 Feb 2022

Dear Dr. Shin,

Thank you very much for re-submitting your manuscript "Association Between First-trimester Exposure to Benzodiazepines and Risk of Congenital Malformations in Offspring: A Population-Based Cohort Study" (PMEDICINE-D-21-04507R2) for consideration at PLOS Medicine.

I have discussed the paper with our academic editor and it was also seen again by two reviewers. I am pleased to tell you that, once the remaining editorial and production issues are fully dealt with, we expect to be able to accept the paper for publication in the journal.

[LINK]

Please let me know if you have any questions, and we look forward to receiving the revised manuscript.   

Sincerely,

Richard Turner, PhD

rturner@plos.org

Requests from Editors:

Regarding the data statement, thank you for including a web address for inquiries about data (https://opendata.hira.or.kr). Readers outside Korea may find this site difficult to navigate, and we ask you to substitute a more specific web address, e.g., https://www.data.go.kr/en/index.do. 

Please remove "Association between" from the title; and add "... in Korea" at the end, or make a similar amendment to indicate where the study was done.

We suggest indicating the end of the follow-up period in the abstract.

At line 41, we suggest amending the wording to "... evidence about possible teratogenicity is equivocal".

At line 46, would "The exposure ..." be preferable to "Intervention"?

At line 48, "confidence intervals", we imagine.

At line 64 (abstract) and any similar instances where you discuss observations, please make that "... was associated". 

At line 78, we suggest "... we found a small increased risk".

Please remove the information about data availability, funding and competing interests from the end of the main text. In the event of publication, this information will appear in the article metadata, via entries in the submission form. A brief "Acknowledgements" section is permissible here, excluding this information.

Throughout the manuscript, including the figures, please substitute "p<0.001" for "p<0.0001" and any other p values less than 0.001. 

Comments from Reviewers:

*** Reviewer #1: 

Thank you to the authors for doing an excellent revision that addressed all my previous comments well. I particularly appreciated the additional sensitivity analysis in appendix S2.

*** Reviewer #2: 

Questions were properly answered

***

[LINK]

---

## [Editor Report · Decision Letter 3]

13 Feb 2022

Dear Dr Shin, 

On behalf of my colleagues and the Academic Editor, Dr Stock, I am pleased to inform you that we have agreed to publish your manuscript "First-trimester Exposure to Benzodiazepines and Risk of Congenital Malformations in Offspring: A Population-Based Cohort Study in South Korea" (PMEDICINE-D-21-04507R3) in PLOS Medicine.

PRESS

Sincerely, 

Richard Turner, PhD 

rturner@plos.org